# Dose-Dependent Blood-Feeding Activity and Ovarian Alterations to PM_2.5_ in *Aedes aegypti*

**DOI:** 10.3390/insects12100948

**Published:** 2021-10-18

**Authors:** Thipruethai Phanitchat, Sumate Ampawong, Artit Yawootti, Thammanitchpol Denpetkul, Napid Wadmanee, Mongkhon Sompornrattanaphan, Chaisith Sivakorn

**Affiliations:** 1Department of Medical Entomology, Faculty of Tropical Medicine, Mahidol University, Bangkok 10400, Thailand; thipruethai.pha@mahidol.ac.th; 2Department of Tropical Pathology, Faculty of Tropical Medicine, Mahidol University, Bangkok 10400, Thailand; am_sumate@hotmail.com; 3Faculty of Engineering, Rajamangala University of Technology LANNA, Chiang Mai 50300, Thailand; YArtit@rmutl.ac.th; 4Department of Social and Environmental Medicine, Faculty of Tropical Medicine, Mahidol University, Bangkok 10400, Thailand; thammanitchpol.den@mahidol.edu; 5Hospital for Tropical Disease, Faculty of Tropical Medicine, Mahidol University, Bangkok 10400, Thailand; napid.wad@mahidol.ac.th; 6Division of Allergy and Clinical Immunology, Department of Medicine, Faculty of Medicine Siriraj Hospital, Mahidol University, Bangkok 10700, Thailand; bankallergymed@gmail.com; 7Department of Clinical Tropical Medicine, Faculty of Tropical Medicine, Mahidol University, Bangkok 10400, Thailand

**Keywords:** *Aedes aegypti*, PM_2.5_, blood-feeding, pollution, arbovirus

## Abstract

**Simple Summary:**

*Aedes aegypti* (*Ae. aegypti*) is a mosquito that transmits arboviruses and responds to various biological and environmental stressors, including temperature, rainfall, and humidity. However, there is a lack of knowledge about fine particulate matter (PM_2.5_) effects on arbovirus vectors. We hypothesized that fine particulate matter (PM_2.5_) may affect *Ae. aegypti* blood-feeding rate and organs. We set up an environmental chamber that could adjust the concentration of PM_2.5_ and observed their blood-feeding activity. We observed a dose–response relationship between PM_2.5_ level and blood-feeding rate in adult female *Ae. aegypti* mosquitoes. In addition, histopathological study showed some changes in the ovaries. Vacuolated or vacuolar degeneration characterized by a formation of non-lipid small droplets in the cytoplasm was observed. This demonstrated the degenerative stage of the cells before developing hydropic degeneration or another advanced stage of cellular damage. The present study explored the effects of PM_2.5_ exposure on the blood-feeding rate and organ integrity in the major arboviral vector *Ae. Aegypti*, providing information on the potential but indirect operational impact of PM_2.5_ exposure on the survival and transmission capabilities of this major vector. Our findings may contribute towards the conceptualization and implementation of mosquito control measures with due consideration of the effects of ambient PM_2.5_ on their populations.

**Abstract:**

High levels of fine particulate matter (PM_2.5_) air pollution are a concern for human health. Several studies have examined the effects of air pollution on human and animal health. However, there is a lack of knowledge about its effects on arbovirus vectors. Thus, we investigated whether PM_2.5_ concentration alters the blood-feeding activity of *Ae. aegypti* mosquitoes. We investigated the effect on the females’ propensity to blood feed at eight concentrations of PM_2.5_ ranging from 100 to 1000 μg/m^3^. Correlation analysis showed blood-feeding activity had a significant strong negative correlation with concentration of PM_2.5_ (*r_p_* = −0.85; *p* ≤ 0.00001). Exploratory linear and non-linear models showed an exponential decay relationship was the best fitting model (corrected Akaike’s information criterion, 193.0; Akaike’s weight, 0.766; adjusted R^2^, 0.780). Ultrastructural study demonstrated PM_2.5_ did not obstruct the respiratory system, but some fine particles were present on the antenna and abdominal body parts. Ovaries showed a dose–response relationship between PM_2.5_ level and vacuolated degeneration. In conclusion, the blood-feeding behavior of *Ae. aegypti* females may have an exponential decay relationship with PM_2.5_ level, and their ovaries may demonstrate dose-dependent degeneration. These findings may be important in understanding the vector’s biology and disease transmission in settings with high PM_2.5_ levels. These results are important to understand blood-feeding and feeding pattern of mosquitoes during PM_2.5_ pollution, which is important for disease transmission and vector control.

## 1. Introduction

Arboviruses (arthropod-borne virus) consist of various groups of viruses normally transmitted by mosquitoes and ticks. They include globally spreading viruses, which cause diseases in humans, such as Zika, dengue, and chikungunya viruses. The term ar-bovirus is not currently part of the viral classification system, which is based on the nature and structure of the viral genome. However, these viruses have similar life-history and transmission patterns, making information gleaned from one virus potentially useful in understanding the others for developing prevention and control measures [1,2]. Dengue is an important public health concern throughout tropical and sub-tropical countries. The dengue virus is a single strand of RNA virus of the Flaviviridae family and is transmitted by the *Aedes* mosquito (primarily the subgenus *Stegomyia*) [3]. In particular, *Ae. aegypti* (L.) is the most prevalent vector in the human–mosquito cycle in tropical and sub-tropical regions, while *Ae. albopictus* (Skuse) is regarded as a secondary vector [3]. Dengue transmission depends on the Aedes population dynamic. Environmental factors show strong effects on *Ae. aegypti* abundance and dengue incidence [4,5]. Understanding the effects of environmental factors will help to target control measures and improve vector control program in endemic areas [6].

Environmental factors, including temperature, rainfall, and humidity, are predictors of dengue infection [7,8,9]. For example, increased temperatures can enhance the transmission potential of the dengue vector [10,11,12,13]. At present, haze and air pollution are problems in many countries, especially in Asia, including Thailand. In Bangkok, annual average ambient particulate matter concentrations of PM_2.5_ exceeded the WHO air quality guideline. The excess fine particulate matter concentrations may pose a potential risk to human health [14]. Haze is an atmospheric phenomenon where dust, smoke, and any particles in the air obscure the sky’s clarity. The World Meteorological Organization (WMO) categorizes atmospheric obscuration by a list of different types. These include: haze, fog, ice fog, steam fog, mist, smoke, volcanic ash, dust, sand, and snow [15]. Particle pollution, or particulate matter (PM), is a mixture of solids and liquid droplets suspended in the air. PM comes in a wide range of sizes, including classifications as coarse (PM_10_), defined as 2.5 to 10 μm in aerodynamic diameter, and fine (PM_2.5_), defined as an aerodynamic diameter ≤2.5 μm [16]. PM_2.5_ consists mainly of combustion particles from motor vehicles, wildfires, and humans burning materials, and it may also contain some crystal particles from finely pulverized road dust and soils [17]. These sources produce particles with different characteristics, and the relative toxicities of these sources and their characteristics are areas of relatively recent but intense interest [18]. There are four main sources of PM_2.5_ in Bangkok and metropolitan areas: automobile exhaust (for which diesel fuel is the main culprit), burning of biomass both indoors and outdoors, secondary dust generated from the combination of automobile exhaust with incomplete combustion, and burning of fossil fuels in factories and electrical generator plants [19].

Not only does PM_2.5_ affect species in the outdoor environment, but it also affects indoor dwellers, as it can enter a building via cracks or the building’s ventilation system [20]. PM_2.5_ can combine with water vapor, smoke, and other gases in the environment. Short-term exposure can exacerbate and worsen the symptoms of chronic respiratory conditions including allergic rhinitis, asthma, and chronic obstructive pulmonary diseases, while long-term exposure may increase the risk of emphysema, lung cancer, and dementia [19,21]. PM_2.5_ affects most human organs and is one of the contributory factors to causing and/or aggravating many respiratory diseases, such as chronic obstructive pulmonary disease, asthma, and lung cancer [22]. In animal models, both the acute and long-term increases in the amount of PM_2.5_ in the air are associated with increased incidence of heart and blood vessel diseases affecting the cardiovascular system and the brain [23]. However, there have been very few studies demonstrating the effect of PM and arthropod vectors in tropical infectious diseases epidemiology [6,24].

The association between haze and dengue incidence was reported in two studies in Singapore. Massad et al. were interested in whether a combination of haze with other local sources of PM had a significant impact on mosquito life expectancy in terms of significantly increasing their mortality rate. Their results showed a lower-than-expected number of dengue cases in Singapore in 2006 was caused by an increase in mosquito mortality due to the above-average haze affecting the country, and they also showed that particularly favorable environmental conditions in 2007 propagated the mosquito population due to a lower mortality rate, which explained the greater-than-expected number of dengue cases in that year [24]. A few years later, Wilder-Smith et al. determined the relationship of dengue activity and haze, which is measured as the pollution standard index (PSI) in Singapore. They found no association between dengue activity and haze [6]. It is notable that Massad et al.’s results were based on mathematical modelling and various assumptions rather than entomological data. Furthermore, Wilder-Smith et al.’s study of the association between air pollutant index as an exposure metric and incidence of dengue as an outcome did not specifically investigate the effects on mosquitoes of PM_2.5_ as an exposure metric. Thus, data on the specific association between PM_2.5_ pollution and the arbovirus vector are sparse, especially data on the specific relationship between PM_2.5_ and the biology of a dengue vector. Thus, we aimed to explore the link between PM_2.5_ exposure and entomological indices of *Ae. aegypti*. A PM_2.5_ generating chamber was set up, and *Ae. aegypti* activity was observed. Histopathological and electron microscopy studies were performed to identify any histopathological or ultrastructural changes on the mosquitoes’ bodies.

## 2. Materials and Methods

### 2.1. Aedes aegypti Populations

*Aedes aegypti* mosquitoes were reared in insectarium at the Department of Medical Entomology, Faculty of Tropical Medicine, Mahidol University, Bangkok, Thailand. The larvae were fed fish food for growth and were developed to the pupa stage. The pupae were harvested into small plastic bowls and stored in a mosquito cage (20 × 20 × 30 cm) for the adults to emerge. The adults were fed with 4% sucrose solution by soaking the solution on a cotton stick.

### 2.2. Environmental Chambers and PM_2.5_ Generator

PM_2.5_ collected with the five-stage and two-stage filter pack air sampler methods by the Pollution Control Department (PCD), and Thailand was characterized [25]. Trace metals were characterized by microwave digestion according to the United States’ Environmental Protection Agency’s (US EPA) method 3051A, which detected the trace metals from PM_2.5_ with inductively coupled plasma optical emission spectroscopy (ICP-OES, Plasma Quant 9000 Elite, Analytik Jena, Jena, Germany).

An experimental chamber was made of acrylic with a volume of 70 × 70 × 70 cm. The photoperiod was 12:12 h of light and darkness daily. Collected PM_2.5_ was dissolved with deionized water and aerosolized into the experimental chamber via atomization technique [26]. An electric fan inside the chamber caused airflow and maintained environmental concentration of PM_2.5_ consistently during the experiment. Control mosquitoes were fed blood in acrylic environmental chambers without PM_2.5_ exposure from the ambient air. Both experimental and control mosquitoes were tested, while environmental factors were controlled. These included a temperature of 26 ± 2 °C and a relative humidity of 50–70%. Accurate measurements of the environmental concentration of PM_2.5_ were performed by DustBoy (www.CMUCCDC.ORG, accessed on 16 October 2021) and UNI-T^TM^ [27,28] (Appendix A). DustBoy is a project of the northern research group in Thailand for monitoring PM_2.5_ in real-time and uses a low-cost light-scattering sensor that provides measurements quickly. The airborne PM is classified by the virtual impactor by removing particles with an aerodynamic diameter >10 μm from the main airflow, thereby allowing only particulate matter with an aerodynamic diameter of ≤10 μm (PM10) to pass through to the sensor. Then, NodeMCU ESP8266 obtains the PM_10_ and PM_2.5_ data from the sensor and records them to a memory card, the data logger. A real-time clock (RTC) is used to generate the recorded date and time. A liquid crystal display is used to show the measurement data. In addition, DustBoy can measure the temperature and relative humidity. DustBoy and a standard PM measuring method (the beta ray attenuation and TeleDyne T640 light-scattering method—US EPA approved) have been compared with PM_2.5_ measurements at standard PCD stations in many areas including Bangkok, Chiangmai, and Ubon Ratchathani provinces) since 2018. The comparison results indicate that DustBoy had a high correlation with the standard PM measuring method [29]. The concentration of PM_2.5_ in an experimental chamber was observed and maintained within the set-up test level by observers who nebulized PM_2.5_ into an experimental chamber according to real-time PM_2.5_ monitoring by DustBoy machine.

### 2.3. Blood-Feeding Activity

Three- to five-day-old females were fed human blood by an artificial feeder system. Membrane feeding assay was preformed using parafilm [30]. The human blood was obtained from the Thai Red Cross Society. For the membrane feeding setting, a circulating water bath was set to 37 °C and connected to holding containers via tubing. A parafilm membrane was stretched across the bottom of the glass tube of the feeder with a surface area of 3.14 cm^2^ and secured with a rubber band. Two mL of blood was added to the funnel of the glass feeder. For each feeding experiment, 100 female mosquitoes were placed in each cage. Following deprivation of sugar solution for 24 h, the cage of females was offered a blood meal. The number of fully blood-fed mosquitoes was counted after the 1 h of feeding. Each set of experiments was conducted in triplicates. The blood-feeding rates in different conditions were calculated as:The blood feeding rate =
(number of blood-fed mosquitoes ÷ number of mosquitoes tested) × 100%

### 2.4. Morphological Study

#### 2.4.1. Scanning Electron Microscopic Study

To distinguish any ultrastructural changes in the whole bodies of *Ae. aegypti* mosquitoes, a scanning electron microscopy (model JSM-6610LV, JEOL, Tokyo, Japan) was used. The mosquitoes were collected from all groups of the experiment (three to five mosquitoes per group) and were keep in −20 °C for 20 to 30 min until no motility was observed, and then they were coated with a sputter coater (EMITECH K550, Emitech Ltd., Ashford, UK). Any fine morphological changes and any other changes were recorded.

### 2.4.2. Histopathological Study

To compare histopathological features between PM_2.5_-exposed and non-PM_2.5_-exposed groups, a histopathological analysis was conducted [27]. The mosquitoes were collected and fixed in 10% neutral buffer formalin for seven days. The specimens were dehydrated with graded ethanol, infiltrated and embedded with paraffin, and cut into 5 µm thickness. After rehydration, the sections were stained with hematoxylin for 15 min and eosin for 2 min, and then they were dehydrated with ethanol and mounted with DePeX™. Histopathological changes in the whole bodies of mosquitoes were examined under a light microscope. Histopathological findings were scored and presented in percentage of abnormal cell/section. In this study, ovarian vacuolated degeneration as characterized by a formation of non-lipid droplets in the cytoplasm of ovarian cells was measured and compared with a total number of these cells.

### 2.5. Statistical Analysis

Quantitative data are presented descriptively. Blood-feeding rates between PM_2.5_ levels were compared by the Kruskal–Wallis test. The correlation between PM_2.5_ level and blood-feeding rate of *Ae. aegypti* was estimated by Spearman’s correlation coefficient. Five regression models coding PM_2.5_ as a categorical variable were fitted to explore the dose–response relationships between PM_2.5_ level and blood-feeding rate: a linear relationship for model 1, a 2-segment piecewise linear relationship for model 2, a 3-segment piecewise linear relationship for model 3, an exponential decay relationship for model 4, and a non-linear relationship by restricted cubic spline with 3 knots for model 5 [31]. Akaike’s Information Criterion (AIC_c_), which corrects for small sample size bias [32], was calculated for each model. The best fitting model was selected by Akaike’s weights and evidence ratios [33]. The Akaike’s weight for each considered model represents the weight of evidence for that model, with a larger weight being stronger evidence. The evidence ratio is the ratio of Akaike’s weights for the best fitting model to another model with a larger evidence ratio, being weaker evidence that the non-best-fitting model by Akaike’s weight is the true best fitting model. Change points for piecewise linear regression were selected by scatterplot inspection. For piecewise linear regression, estimated coefficients for each segment’s slope and the significance of change points between segments were tested by the Wald test. Analysis was performed using R version 3.6.1 (R Foundation for Statistical Computing, Vienna, Austria). A *p*-value of <0.05 was considered significant.

## 3. Results

### 3.1. Blood-Feeding Activity

One hundred female mosquitoes were allowed to blood feed while exposed to various concentrations of PM_2.5_ in the experimental chamber, ranging from 50 to >1000 µg/m^3^. Control mosquitoes were fed blood in an acrylic environmental chamber with PM_2.5_ in ambient air with concentration 0–5 µg/m^3^, and their blood-feeding rate was >90%. There was a significant difference in the effect of PM_2.5_ on mosquito blood-feeding of *Ae. aegypti* by Kruskal–Wallis test (*p* = 0.013). When PM_2.5_ was generated and circulated in the chamber, blood-feeding rate decreased significantly. At a PM_2.5_ level of 50–100 µg/m^3^, the blood-feeding rate ± SD (range) was 37.3% ± 18.3 (23.0–58.0), which was a decrease of 41.0% compared to the controls. At a PM_2.5_ level of 550–700 µg/m^3^, the blood-feeding rate was <20% with a minimum of 7% (Table 1). There was a significant negative correlation between concentrations of PM_2.5_ and blood-feeding rate (r_p_ = −0.85; *p* ≤ 0.00001).

Among the five plausible candidate models of the dose–response relationship investigated, the non-linear exponential decay model was the best fitting by lowest AIC_c_ (AIC_c_, 193.0) and Akaike’s weight (Akaike’s weight, 0.766) (Table 2 and Figure 1). The non-linear exponential decay model also had the highest adjusted R^2^ value among the five models (adj R^2^, 0.780). The second-best-fitting model was the two-segment piecewise linear model with an evidence ratio of 5.21, suggesting that the apparent best fitting model (the exponential decay one) has quite strong support as the true best fitting model (Table 2 and Figure 2). Among the piecewise linear regression models, the Wald tests for the two- and three-segment models were significant for the second segments (*p* = 0.003 and 0.03, respectively). However, the Wald test for the third segment in the three-segment model was not significant (*p* = 0.536). The restricted cubic spline, three-segment piecewise linear model, and the linear model had evidence ratios suggesting much poorer fits to the data than the non-linear exponential decay model (Table 2 and Appendix A).

### 3.2. Histopathological and Morphological Analysis

Ultrastructural images confirm that the sizes of particles classified as PM_2.5_ obtained from the PCD were <2.5 μm (Figure 3). After dissolving the PM_2.5_ particles into distilled water, the PM_2.5_ was visualized more clearly than on the filter paper. In terms of morphological characteristics, the particles were mixed round and rod shapes with smooth surfaces (Figure 3A,B). Under experimental PM_2.5_ levels of exposure, many fine particles were attached to the mosquitoes’ bodies, while no particles were seen on the control mosquitoes (Figure 4). The appearance of the PM_2.5_ present on the mosquitoes’ bodies was similar to that generated by environmental chambers. Scanning electron microscope study showed no obstruction by PM_2.5_ particles around the mosquitoes’ spiracles (Figure 4). However, we found some fine particles presented on the antenna and bodies (Figure 5). Histopathological study showed an increase in vacuolated degeneration of the ovaries of the female *Ae. aegypti* with or without blood-feeding in those exposed to higher concentrations of PM_2.5_ (Figure 6). The metal concentrations obtained from collected PM_2.5_ are displayed in Appendix A.

## 4. Discussion

The arbovirus transmission cycle starts with a susceptible mosquito acquiring a viral infection after it has taken an infected blood meal. When the infected blood arrives at the mosquito’s midgut, the virus binds to the cellular surface of the midgut epithelium and replicates. Then, the virus goes to the hemocoel and is disseminated to secondary tissues, including the salivary glands. The virus transmits to a new host during the next feeding. The female dengue vector requires protein from blood-feeding for egg maturation.

The evolution of blood-feeding in arthropods is well recognized as an adaptation fraught with challenges in finding hosts [34]. The olfactory system enables mosquitoes to locate a host from which they obtain blood meals [35]. In *Aedes* mosquitoes, olfactory receptor expression is localized to the antennae, maxillary palps, and proboscis. In the present study, we found many fine particles on mouth parts and body, especially on antennae. The fine particles adhered to the olfactory receptor area, which may reduce the capacity of a mosquito to find a host. In the present study, blood-feeding results confirm higher levels of PM_2.5_ exposure reduced blood-feeding activity of mosquitoes. Although we can assume that the fine particles did not enter the respiratory tract, they covered the body. This may also create environmental stress for the mosquito. The exponential decay model of the dose–response relationship between PM_2.5_ level and blood-feeding rate was the best fitting model in the present study, evidenced by Akaike’s weight and evidence ratio. However, this claim about the best fitting dose–response model needs validation in future studies.

In the present study, histological study demonstrated an increase in vacuolated degeneration of the ovaries in female *Ae. aegypti* exposed to higher concentrations of PM_2.5_ without any changes in morphological features or obstruction of the spiracles in female *Ae. aegypti* exposed to PM_2.5_. The observed vacuolated degeneration of the ovaries needs further longitudinal study to see the effect on the reproductive or life cycle of *Ae. aegypti*. In addition, it has been reported that oogenesis maturation is blocked in non-blood-fed female *Ae. aegypti* [36] Ovarian apoptosis and its associated lesions may be the important causes of oval growth retardation in mosquitoes [37]. The present study demonstrated the ovaries were intact in negative control mosquitoes with blood meal, while those without blood-feeding showed a mild degree of ovarian degeneration. When the mosquitoes were exposed to PM_2.5_ at a level of more than 100–150 µg/m^3^ both with and without blood meal, mosquitoes also presented with this lesion in association with a dose–response trend. We postulate fine particulate exposure may lead to ovarian defects independently of blood-feeding. However, mechanistic details need to be confirmed with further studies.

In Thailand, the levels of PM_2.5_ range from lower than 10 µg/m^3^ to >200 µg/m^3^ [38]. The presence and continuously increasing levels of these tiny particles have been a worsening issue in Thailand over the past few years, with regular cycles of safe to unsafe levels for human health reported weekly [38]. The increase in PM_2.5_ not only contributes to adverse health outcomes in humans, but it may also have a benefit by reducing the risk of *Ae. aegypti* mosquito-borne diseases, including dengue, chikungunya, and Zika disease, because of the strong negative correlation between PM_2.5_ level and blood-feeding activities of *Ae. aegypti* observed in the present study.

A limitation of the study is the time of observation in mosquitoes after exposure to PM_2.5_. It should be increased in future studies to investigate the longitudinal effects on the ovary cells.

This study is the first report to demonstrate fine particle as a stressor associated with reduced blood-feeding activity in *Ae. aegypti* mosquitoes. High concentrations of fine particles in the air may reduce the function of olfactory receptors and the ability to find a host. PM_2.5_ pollution is a temporary stressor. However, its concentration is influenced by complex meteorological factors and human activities [39]. The results are important to understand blood-feeding or feeding patterns of mosquitoes during PM_2.5_ pollution. This information has the potential to inform the accurate conceptualization and implementation of successful control programs during high levels of PM_2.5_. Further longitudinal studies are needed to see the effect of PM_2.5_ on the life cycle of *Ae. aegypti* and to confirm the association between air pollution and vector-borne diseases.

## Figures and Tables

**Figure 1 insects-12-00948-f001:**
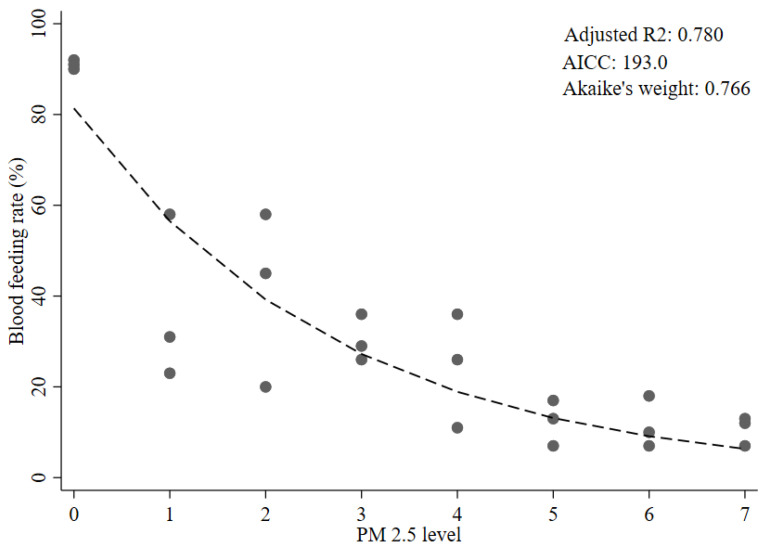
Exponential decay model of the dose–response relationship between PM_2.5_ level and blood-feeding rate in female *Ae.aegypti* mosquitoes.

**Figure 2 insects-12-00948-f002:**
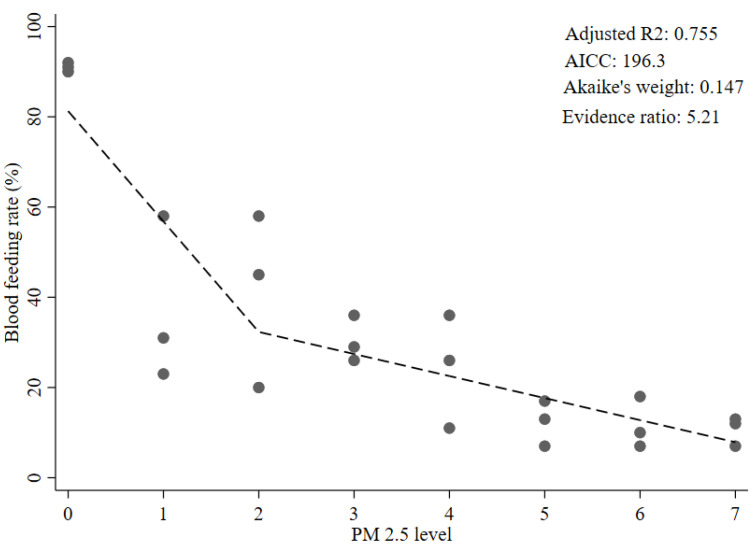
Two-segment piecewise linear model of the dose–response relationship between PM_2.5_ level and blood-feeding rate in female *Ae. aegypti* mosquitoes.

**Figure 3 insects-12-00948-f003:**
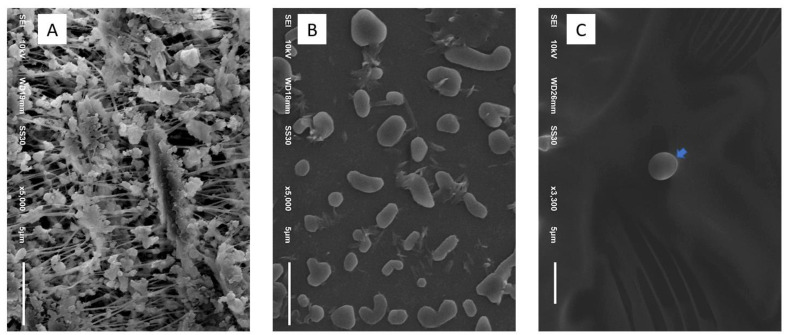
Appearance of PM_2.5_ by scanning electron microscopy from filter paper (**A**), PM_2.5_ extracted solution (**B**), and the abdominal body parts of a mosquito (**C**).

**Figure 4 insects-12-00948-f004:**
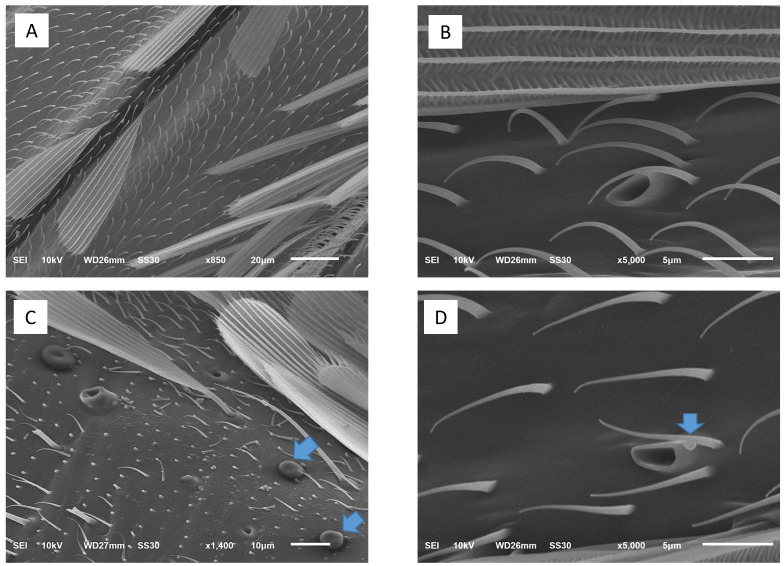
Ultrastructures of the mosquitoes with and without PM_2.5_ exposure: (**A**,**B**) a negative control mosquito showing no particles on the mosquito’s body and no obstruction around the spiracles; (**C**,**D**) a PM_2.5_-exposed mosquito showing particles on the body and no obstruction around the spiracles.

**Figure 5 insects-12-00948-f005:**
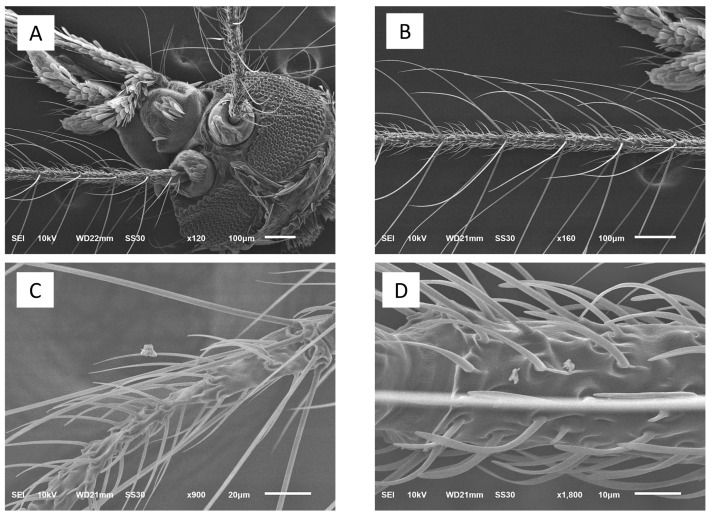
Fine morphology of the antenna of mosquitoes with and without PM_2.5_ exposure: (**A**,**B**) a negative control mosquito showing no particles on the antenna; (**C**,**D**) a PM_2.5_-exposed mosquito showing particles on the antenna.

**Figure 6 insects-12-00948-f006:**
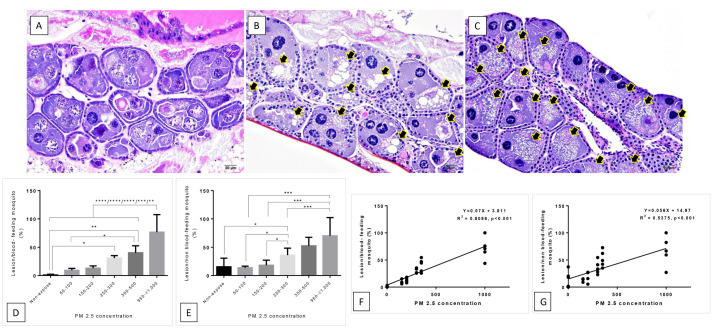
Ovarian histopathological change and its score from the female *Ae. aegypti* with or without PM_2.5_ exposure in accordance with the correlation between the pathological severity and the concentration of PM_2.5_: comparing with negative control group as demonstrated by intact ovarian cells (**A**), the results show an increase in vacuolated degeneration (arrow) of the ovaries in both female *Ae. aegypti* with (**B**) and without (**C**) blood-feeding at the highest concentration (>1000 µg/m^3^). The pathological severity tends to increase as concentrations of PM_2.5_ increase in the mosquitoes both with and without blood-feeding (**D**–**G**) (*: *p* < 0.05, **: *p* < 0.01, ***: *p* < 0.001, and ****: *p* < 0.0001), with a positive correlation at R^2^ = 0.8086, *p* < 0.001, and R^2^ = 0.5375, *p* < 0.001, respectively.

**Table 1 insects-12-00948-t001:** Blood-feeding rates in female *Ae. aegypti* mosquitoes stratified by PM_2.5_ concentration.

Conc. of PM_2.5_ (µg/m^3^)	PM_2.5_ Level Category	Total Number of Mosquitoes	Blood-Feeding Rate, %, Mean ± SD (Range)
^a^ 0–5	0	100	91.0 ± 1.00 (90, 92)
50–100	1	100	37.3 ± 18.3 (23, 58)
150–200	2	100	41.0 ± 19.3 (20, 58)
250–300	3	100	30.3 ± 5.1 (26, 36)
350–500	4	100	24.3 ± 12.6 (11, 36)
550–700	5	100	12.3 ± 5.0 (7, 17)
750–900	6	100	11.7 ± 5.7 (7, 18)
950–≥1000	7	100	10.7 ± 3.2 (7, 13)

Notes: ^a^ PM_2.5_ level 0–5 µg/m^3^ was the control group. Abbreviations: PM_2.5_, particulate matter with an aerodynamic diameter <2.5 µm; SD, standard deviation.

**Table 2 insects-12-00948-t002:** Linear and non-linear regression models of the relationship between PM_2.5_ level and blood-feeding rate in female *Ae. aegypti* mosquitoes.

		PM_2.5_ Level Category	β	SE	*p*-Value	Adjusted R^2^	AIC_c_	^a^ ΔAIC_c_	Akaike’s Weight [26]	^b^ Evidence Ratio
**Linear**										
**^b^ Model 1**	B0		64.9			0.642	203.4	10.4	0.004	191.5
	X1		−9.32	1.43	<0.0001					
**Piecewise linear**										
**^c^ Model 2** **(2 segments)**	B0		81.3			0.755	196.3	3.3	0.147	5.21
	X1	0–1	−24.5	4.68	<0.0001					
	X2	2–7	−4.89	1.78	0.012					
**^d^ Model 3** **(3 segments)**	B0		81.0			0.747	198.9	5.9	0.042	18.2
	X1	0–1	−23.5	5.06	<0.0001					
	X2	2–4	−6.32	3.22	0.064					
	X3	5–7	−2.34	5.06	0.649					
**Non-linear exponential decay**										
**^e^ Model 4**	B0		81.3	6.46	<0.0001	0.780	193.0	Ref.	0.766	Ref.
	B1		−0.365	0.0509	<0.0001					
**Restricted cubic spline**										
**^f^ Model 5**	B0		76.5			0.729	198.8	5.8	0.042	18.2
	Spline 1		−17.7	3.19	<0.0001					
	Spline 2		11.2	3.91	0.010					

Notes: The Wald test *p*-values comparing difference in slopes between segments in piecewise linear models were *p* = 0.003 for model 2, *p* = 0.03 for segment 1 vs. segment 2 in model 3, and *p* = 0.596 for segment 2 vs. segment 3 in model 3. A *p*-value < 0.05 was considered significant. Abbreviations: AIC_c_, corrected Akaike’s information criterion; CI, confidence interval; PM_2.5_, particulate matter with an aerodynamic diameter < 2.5µm; SE, standard error. ^a^ The difference in AIC_c_ between the best fitting model (Ref. = lowest AIC_c_ and best fitting model) and the model. ^b^ The evidence ratio was calculated as the Akaike’s weight of the best fitting model by AIC_c_ (lowest AIC_c_) divided by the Akaike’s weight of the model of interest. ^b^ Model 1: Y = B0 + B1 × 1. ^c^ Model 2: Y = B0 + B1 × 1 + B2 × 2. The beta coefficients B1 and B2 represent the slopes of each segment. ^d^ Model 3: Y = B0 + B1 × 1 + B2 × 2 + B3 × 3. The beta coefficients B1 to B3 represent the slopes of each segment. ^e^ Model 4: Y = B0e^B1×1^. ^f^ Model 5: Y = B0 + Spline 1 + Spline 2. Restricted cubic spline fitting was performed with 3 knots at the 10th, 50th, and 90th percentiles of PM2.5 level [31].

## Data Availability

All data are contained in the present article.

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
