# Peer review of "Dose-Dependent Blood-Feeding Activity and Ovarian Alterations to PM2.5 in Aedes aegypti"

_insects, 2021, doi:10.3390/insects12100948_

Round 1

Reviewer 1 Report

Dear authors,

This paper deals with the effect of fine particulate matter (PM2.5) air pollution on Ae. aegypti blood feeding behavior including the evaluation of morphological and histological effects. The results are quite interesting since a negative dose response relationship between PM2.5 level and blood feeding activity was recorded and a vacuolated degeneration in the ovary.

Further down, you will find some comments of minor importance that may benefit the manuscript.

Lines 91-92: You may add the references from the few relevant studies you are mentioning.

Lines 108-111: I propose a slight rewording to just state the scope of the manuscript, not the methodology: “Histopathological and electron microscopy studies were performed to identify the fine particles to the mosquito body and any morphological changes in the respiratory system the ovaries of Ae. aegypti”.

Table 1: I suggest removing columns with “blood feeding rate by replicate (%)” as redundant information. Please check the PM2.5 concentration in the control group in the table and the footnote. 5-25 or 0-25 μg/m3?

In the discussion you could discuss in few lines the results concerning the best fitting model (non-linear exponential decay model) and the potential to use it as a predictor to blood feeding activity of Ae.aegypti in PM2.5 polluted areas.

Lines 337-355: This paragraph provides too much information about the effects of environmental abiotic factors, such as temperature and food, on biological traits of mosquitoes and disease transmission. However, this seems not so relevant to the topic of the article which is the effects of air pollutants to mosquitoes. Hence, this paragraph should be limited or even removed. Instead, data from relevant studies with air pollutants and mosquitoes or other arthropods could be discussed here.    

Author Response

Reviewer 1

Lines 91-92: You may add the references from the few relevant studies you are mentioning.

Response: Done.

Lines 108-111: I propose a slight rewording to just state the scope of the manuscript, not the methodology: “Histopathological and electron microscopy studies were performed to identify the fine particles to the mosquito body and any morphological changes in the respiratory system the ovaries of Ae. aegypti”.

Response: Done. 

Table 1: I suggest removing columns with “blood feeding rate by replicate (%)” as redundant information. Please check the PM2.5 concentration in the control group in the table and the footnote. 5-25 or 0-25 μg/m3?

Response: Done.   

In the discussion you could discuss in few lines the results concerning the best fitting model (non-linear exponential decay model) and the potential to use it as a predictor to blood feeding activity of Ae.aegypti in PM2.5 polluted areas.

Response: Done.

Lines 337-355: This paragraph provides too much information about the effects of environmental abiotic factors, such as temperature and food, on biological traits of mosquitoes and disease transmission. However, this seems not so relevant to the topic of the article which is the effects of air pollutants to mosquitoes. Hence, this paragraph should be limited or even removed. Instead, data from relevant studies with air pollutants and mosquitoes or other arthropods could be discussed here. 

Response: Done.

Reviewer 2 Report

Dear Authors,

I have read through your manuscript and have made corrections which can make it better. Mostly typographical errors and grammar. However, my major concern is there is no real control in your study. By exposing your control to 5-25 μg/m3 of PM2.5 make them just another set of experimental lines for lower concentration. This cannot be a representative control, to provide unbiased comparator for your blood feeding assessment, as well as morphological and histopathological analyses. 

A proper control is required. 

Author Response

Reviewer 2

I have read through your manuscript and have made corrections which can make it better. Mostly typographical errors and grammar. However, my major concern is there is no real control in your study. By exposing your control to 5-25 μg/m3 of PM2.5 make them just another set of experimental lines for lower concentration. This cannot be a representative control, to provide unbiased comparator for your blood feeding assessment, as well as morphological and histopathological analyses.

A proper control is required.

Response: Thank you very much for your comment, I agree with your suggestion. In this study, we try to eliminate all PM2.5 by using environmental chambers together with air purifier during experiment of control. However, it was present some PM2.5 between 0-5 µg/m3 which the concentration of PM 2.5 mostly close to 0 µg/m3. We used this condition as a control because blood feeding rate was more than 90% which mean mosquito was suitable and active for this experimental environment. We wrote PM2.5 between 0-25 µg/m3 because it is the same level of PM 2.5 in an ambient of general environment. We did not applied any PM2.5 in the control (https://www.who.int/data/gho/data/themes/air-pollution/ambient-air-pollution).

Line 22: We hypothesized that fine…

Response: Done.

Line 23/24: I will suggest changing into something like this: ‘Exposure of adult, female Ae. aegypti to particulate matter (PM2.5), resulted in a dose-dependent response between the PM2.5 levels and blood feeding rate.’ Indicate the stage of mosquitoes, their sex, etc, if both males and females used, no need, but important to say larvae, pupae or adults, and if possible, even the age.

Response: Done.

Line 26: Please delete ‘To the best of our knowledge’

Response: Done.

Line 27: what outcomes? Please say it.

Response: Done.

Line 28: during transmission of what? Better to say ‘the present study explores the effects of PM2.5 exposure on the blood feeding rate and organs integrity in the major arboviral vector Ae. Aegypti, providing information on the potential, but indirect operational impact of PM2.5 exposure on survival and transmission capabilities of this major vector.’

Response: Done.

Line 29: It looks like authors are proposing that Ae. aegypti should be protected? Simple summary fails to capture the essence of this work. Authors need find a balance between the role of pollution in the environment and its potential to crash down the populations of mosquito vectors.

Response: Thank you very much for your suggestion.

Abstract

Line 32: Thus, we investigated whether PM2…

Response: Done.

Line 40: What body parts, exactly?

Response: Abdomen part and focus on ovaries.

Line 41:…female Ae. aegypti may have. There is a period missing after Ae

Response: Done.

Introduction

Line 61: Provide more references if available, please.

Response: Done.

Line 62: correct en-viron-mental

Response: Done.

Line 72: correct crustal

Response: Done.

Lines 70-72: Too weighty without references. Support this with references.

Response: Done.

Line 77: fix automo-bile

Response: Done.

Lines 94 and 97: fix in-creas-ing and environ-mental  

Response: Done.

Line 106: is better to say link between PM exposure and transmission or entomological indices.

Response: Done.

Lines 107-110: Please remove description of methods. Rather dwell on what you have found. Like here we found a clear relationship between exposure and concentrations of PM and reduction in blood feeding, as well as degeneration in the ovaries, findings which are of significance in terms of viral transmission by the Ae. aegypti.

Response: Done.

Materials and Methods

Line 112: remove colonization

Response: Done.

Line 114: name should be in full, Aedes aegypti, in the beginning of the sentence.

Response: Done.

Line 120: For non-experts this description is not understandable at all. In which form is this PM25. Authors should endeavor to explain this clearly, as well as the PM2.5 generator. What does that mean?

Response: We have given the methods and references of how PM2.5 were collected and aerosolized into the experimental chamber. (Page 3)

Line 127-128: please delete ‘PM2.5 was collected by the PCD, Thailand.’ Authors have said that already.

Response: Done.

Line 133: is this temperature not too high to stress the mosquitoes?

Response: I am very sorry, it was not correct. This study temperature was set up at 26±2°C.

Line 135: Figure S1

Response: Done.

Line 130: Authors said the control mosquitoes were fed blood and with PM2.5 exposure, which ones are experimental then? And sentence need corrections fed blood, not feed blood.

Response: Done. (Change to without PM2.5)

Line 143: correct liq-uid and measurement data?

Response: Done.

Line 150: ‘Three- to five-day old…’ and which artificial feeding system use. Say it please.

Response: Done.

Lines 154/155: mL and was added not were added.

Response: Done.

Line 159: ‘The blood feeding rate…’

Response: Done.

Line 164: Authors should delete Morphological and histopathological analysis. Modify line 165 to Morphological Study.

Response: Done.

Line 179: statistical analysis is too long. Reduce to half this volume, please.

Response: Done.

Results:

Line 214: is better to say like ‘…were allowed to blood feed while exposed to various concentrations of PM2.5, ranging from 50 to >1000 ug/m3.’

Response: Done.

Major

Line 216: I cannot fathom the reason why the control mosquitoes will also be exposed to 5-25ug/m3 of the PM2.5. This meant authors have no real control in this study and this paper cannot be published in its present form. Authors ought to add 25ug/m3 to their experimental cohorts and blood feed unexposed females, of the same age, under the same condition, but without any PM2.5 exposure.

Response: Thank you very much for your comment, I totally agree with you suggestion. We explain in above.

Reviewer 3 Report

With air pollution a growing issue in many parts of the world, and an increasing burden of mosquito-borne disease in urban areas, this study into the effect of fine particulate pollution in the atmosphere on blood feeding activity and fertility is a timely and interesting one.

There are some minor points of clarification and additional detail which would make the paper clearer, described below. But the major issue with the paper is that the justification for the study design and the significance and implication of the results is not clear enough. A dramatic impact of particulate air pollution on blood feeding response is presented and this is a novel and interesting result. I would like to see a bit more discussion of how this might affect disease transmission in a real life setting (Bangkok for example) where air pollution may be worsening. I think this part of the study alone would make an interesting paper.  What is much less clear to me is the significance of the morphological and histological part of the study – I do not really understand the methods that were used, what the results show, or what the significance or impact might be of this part. At the moment, without further experiments to look at fertility of exposed females the histological section doesn’t feel like it adds much, and the paper would be simpler if it just showed the effects on blood feeding.

  • The term vacuolated degeneration is not defined or described anywhere, but it is not necessarily a familiar term for all readers. It should probably not be used in the Simple Summary, and even in the Abstract I would recommend a definition.
  • In the Results I would like to see a description of what morphology you look for to identify this phenotype, in the Discussion I would like to see information about the implication of an increased proportion of vacuolated ovaries, and perhaps also a justification of why it was scored in the Methods section.
  • Figure 6 shows a lot of images but without much description of what they show – the arrows are hard to see and it is hard to know how to compare the different treatments. Might it be better to replace this with an example photo and then give values for the proportion of ovarioles which have undergone vacuolated degeneration in each treatment so that the relationship with concentration of PM can be seen?
  • Similarly, there are some images in Figures 3-5 of particles on surfaces and mosquito exteriors, but it is not clear what the significance of these is – what effect might the particles have on mosquitoes, how do they relate to changed blood feeding or other changes, is there a relationship between number of particles and the concentration of PM in the atmosphere?

Specific Comments:

There are a lot of technical terms in the Simple Summary, and the experiment is not as well described as in the Abstract.

The last sentence of the Abstract does not give any firm conclusion – I would prefer a more decisive statement about the possible implications of the results of this study.

The Introduction could be strengthened by giving a clearer justification of why this study was done – it might be easier to follow if it starts with discussion of sources of air pollution in Bangkok (is it a growing problem specifically in this city?), the types of air pollution (why did you decide just to look at PM2.5?) and then the possible effects on mosquitoes and disease transmission (what is known already and what did you set out to find out?). It would also be helpful to add a sentence to the Introduction to explain why you might expect PM to affect blood feeding behaviour and histopathology. The description of the study in the Introduction mentions looking at the respiratory system but not the ovaries, and a better description of what questions you set out to answer and what experiments were done would be helpful.

Please define ‘haze’ or replace the word with a more specific term. Lines 104-105 states there have been ‘no studies investigating the direct relationship between PM and a dengue vector’ but 2 studies have been quoted which looked at the effect of haze – being more specific about the difference or specific definitions would make this claim clearer.

The last paragraph of the Introduction describes one study showing an effect of haze on dengue transmission in Singapore, and then a second which found no effect. It would be useful to try and explore why they might have differed, and how this current study might help to understand the relationship better.

Methods and materials:

  • it is not clear how the PM2.5 were made airborne (aerosolised?)
  • It is not clear that a range of concentrations were tested, or how the concentrations were established, measured or maintained.
  • It would be useful to know what tissues you looked at for morphological changes, and what changes you looked for.
  • What do hematoxylin and eosin stain? A citation for ‘standard tissue processing’ and staining would be useful if the details are not given.

The first sentence of the Results needs to make it clear that the PM is in the atmosphere, not in the blood as it currently reads.

Figure 3 – what body part is shown in image C?

In the Discussion, what is the evidence for the claim that ‘we can assume that the fine particles did not enter the respiratory tract’? Similarly, it is not clear from the results presented that there was no changes in morphological features.

Citation needed for the statement ‘in Thailand, the levels of PM2.5 range from lower than 10 ug/m3 to >200 ug/m3’. More discussion of the effect this level, or the expected level in the future, might have on mosquitoes and on disease transmission would be helpful. This section might be better as an opening paragraph for the Discussion to describe the background to why the study was done.

The following sentence is an important one, but is not very clear and it would be interesting to expand and give more detail: ‘The importance of this response may be targeted to disrupt in disease vector’.

The last paragraph of the Discussion introduces a lot of concepts and is a bit confused. It would be useful to expand this section to give more detail, particularly into the last sentence about how the results of the study could be used to improve vector control in a setting of increasing air pollution.

Close editing for English is needed to correct a number of small errors, for example hyphens where they are not needed, species names in italics, and subscript in PM2.5.

Author Response

Reviewer 3

I would like to see a bit more discussion of how this might affect disease transmission in a real life setting (Bangkok for example) where air pollution may be worsening. I think this part of the study alone would make an interesting paper.  What is much less clear to me is the significance of the morphological and histological part of the study.

Response: Thank you very much for your suggestion.

The term vacuolated degeneration is not defined or described anywhere, but it is not necessarily a familiar term for all readers. It should probably not be used in the Simple Summary, and even in the Abstract I would recommend a definition.

Response: Thank you for your comment. Actually, vacuolated or vacuolar degeneration is a common term using in histopathological filed especially in mammal tissue which characterized by a formation of non-lipid small droplet in the cytoplasm. This term demonstrates the degenerative stage of the cell before develop to hydropic degeneration or another advance stage of cellular damage. Therefore, for non-familiar person, a brief explanation was added on Simple Summary part. Please kindly find.

In the Results I would like to see a description of what morphology you look for to identify this phenotype, in the Discussion I would like to see information about the implication of an increased proportion of vacuolated ovaries, and perhaps also a justification of why it was scored in the Methods section.

Response: Thank you very much for your comment. An explanation term was added the Simple Summary part. This study reported an increasing trend in vacuolated degeneration of ovaries by histopathological observation. We qualitatively reported by morphological change and without score.

Figure 6 shows a lot of images but without much description of what they show – the arrows are hard to see and it is hard to know how to compare the different treatments. Might it be better to replace this with an example photo and then give values for the proportion of ovarioles which have undergone vacuolated degeneration in each treatment so that the relationship with concentration of PM can be seen?

Response: Done.

Similarly, there are some images in Figures 3-5 of particles on surfaces and mosquito exteriors, but it is not clear what the significance of these is – what effect might the particles have on mosquitoes, how do they relate to changed blood feeding or other changes, is there a relationship between number of particles and the concentration of PM in the atmosphere?

Response: Figures 3-5 do not explain all how do they relate to change blood feeding or other changes but they may assume that during environmental stress with PM2.5 effect to blood feeding activity. The figures showed particles on mosquito proof that particles of PM2.5 from generator not only on the air but also attach to the mosquito body and may increase stress to mosquito.

Specific Comments:

There are a lot of technical terms in the Simple Summary, and the experiment is not as well described as in the Abstract.

Response: Done.

The last sentence of the Abstract does not give any firm conclusion – I would prefer a more decisive statement about the possible implications of the results of this study.

Response: Done.

The Introduction could be strengthened by giving a clearer justification of why this study was done – it might be easier to follow if it starts with discussion of sources of air pollution in Bangkok (is it a growing problem specifically in this city?), the types of air pollution (why did you decide just to look at PM2.5?) and then the possible effects on mosquitoes and disease transmission (what is known already and what did you set out to find out?). It would also be helpful to add a sentence to the Introduction to explain why you might expect PM to affect blood feeding behaviour and histopathology. The description of the study in the Introduction mentions looking at the respiratory system but not the ovaries, and a better description of what questions you set out to answer and what experiments were done would be helpful.

Response: Done.

Please define ‘haze’ or replace the word with a more specific term. Lines 104-105 states there have been ‘no studies investigating the direct relationship between PM and a dengue vector’ but 2 studies have been quoted which looked at the effect of haze – being more specific about the difference or specific definitions would make this claim clearer.

Response: Done.

The last paragraph of the Introduction describes one study showing an effect of haze on dengue transmission in Singapore, and then a second which found no effect. It would be useful to try and explore why they might have differed, and how this current study might help to understand the relationship better.

Response: Done.

Methods and materials:

it is not clear how the PM2.5 were made airborne (aerosolised?)

Response: We have given the methods and references of how Pm2.5 were collected and aerosolized into the experimental chamber. Please kindly find on Environmental chambers and PM2.5 generator part.

It is not clear that a range of concentrations were tested, or how the concentrations were established, measured or maintained.

Response: PM2.5 was produced by generator in environmental chamber and measured by DustBoy. The concentration was observed when concentration up to the point at we want then closed the generator, if concentration drop will open the generator.

It would be useful to know what tissues you looked at for morphological changes, and what changes you looked for.

Response: We are looking all tissues of mosquito however we found only changing at the ovary tissues.  

What do hematoxylin and eosin stain? A citation for ‘standard tissue processing’ and staining would be useful if the details are not given.

Response: Thank you for your suggestion. We performed H&E staining and standard tissue processing which are basic techniques in histopathological field. However, brief explanation was added following your recommendation. Please kindly find on page 4.

The first sentence of the Results needs to make it clear that the PM is in the atmosphere, not in the blood as it currently reads.

Response: Done.

Figure 3 – what body part is shown in image C?

Response: Abdomen.

In the Discussion, what is the evidence for the claim that ‘we can assume that the fine particles did not enter the respiratory tract’? Similarly, it is not clear from the results presented that there was no changes in morphological features.

Response: We could not find any fine particle obstruction or presented in the respiratory tract. According of both scanning electron microscopic study and histopathological study were no changes in morphological features.

Citation needed for the statement ‘in Thailand, the levels of PM2.5 range from lower than 10 ug/m3 to >200 ug/m3’. More discussion of the effect this level, or the expected level in the future, might have on mosquitoes and on disease transmission would be helpful. This section might be better as an opening paragraph for the Discussion to describe the background to why the study was done.

Response: Done.

The following sentence is an important one, but is not very clear and it would be interesting to expand and give more detail: ‘The importance of this response may be targeted to disrupt in disease vector’.

Response: This part has been deleted.

The last paragraph of the Discussion introduces a lot of concepts and is a bit confused. It would be useful to expand this section to give more detail, particularly into the last sentence about how the results of the study could be used to improve vector control in a setting of increasing air pollution.

Response: Done.

Close editing for English is needed to correct a number of small errors, for example hyphens where they are not needed, species names in italics, and subscript in PM2.5.

Response: Done.

Round 2

Reviewer 2 Report

Dear Authors,

Thank you for revising this manuscript and providing explanations where concerns are raised. Manuscript is now better. Congrats. 

Author Response

Thank you for your valuable comments that helped us to improve our manuscript. We hope to get your help again and to collaborate with you in the future.

Reviewer 3 Report

Some changes have been made to the manuscript which have improved its clarity and helped the reader to understand the methods and results more easily. However, I would still like to see better framing of the study – why these were interesting questions, what the observations of histopathology and morphology mean, what the implications of the data are, and how the results might be used to better understand disease transmission or improve mosquito control.

The following changes have not been made in the text, even though in some cases the explanations are given in the authors’ response:

Figure 6 shows a lot of images but without much description of what they show – the arrows are hard to see and it is hard to know how to compare the different treatments. Might it be better to replace this with an example photo and then give values for the proportion of ovarioles which have undergone vacuolated degeneration in each treatment so that the relationship with concentration of PM can be seen?

Response: Done.

Reviewer: I think that the author has said that this was a qualitative assessment of the presence of vacuolated ovaries, and so it is not appropriate to quantify the number in the way I suggested.

Similarly, there are some images in Figures 3-5 of particles on surfaces and mosquito exteriors, but it is not clear what the significance of these is – what effect might the particles have on mosquitoes, how do they relate to changed blood feeding or other changes, is there a relationship between number of particles and the concentration of PM in the atmosphere?

Response: Figures 3-5 do not explain all how do they relate to change blood feeding or other changes but they may assume that during environmental stress with PM2.5 effect to blood feeding activity. The figures showed particles on mosquito proof that particles of PM2.5 from generator not only on the air but also attach to the mosquito body and may increase stress to mosquito.

Reviewer: No text has been added to the manuscript to explain why the ovaries were examined, or why the presence of particles on the mosquitoes’ surface was observed. This explanation given here would be useful in the text, along with a mention that the number of particles and vacuolated ovaries increased with PM2.5 concentration, and that the mosquitoes were likely under more stress as a result.

 It would also be helpful to add a sentence to the Introduction to explain why you might expect PM to affect blood feeding behaviour and histopathology. The description of the study in the Introduction mentions looking at the respiratory system but not the ovaries, and a better description of what questions you set out to answer and what experiments were done would be helpful.

Response: Done.

Reviewer: Some text has been added to the Introduction, but it still does not explain why these parameters were looked at, and what the implication might be.

It is not clear that a range of concentrations were tested, or how the concentrations were established, measured or maintained.

Response: PM2.5 was produced by generator in environmental chamber and measured by DustBoy. The concentration was observed when concentration up to the point at we want then closed the generator, if concentration drop will open the generator.

Reviewer: This explanation should be added to the manuscript – it is still not clear as written that a range of concentrations were used or how this was achieved.

It would be useful to know what tissues you looked at for morphological changes, and what changes you looked for.

Response: We are looking all tissues of mosquito however we found only changing at the ovary tissues.

Reviewer: This should be stated in the manuscript.

The last paragraph of the Discussion introduces a lot of concepts and is a bit confused. It would be useful to expand this section to give more detail, particularly into the last sentence about how the results of the study could be used to improve vector control in a setting of increasing air pollution.

Response: Done.

Reviewer: One sentence was added to restate what was done in the study, and I would still like to see more discussion of the possible usefulness of the data collection and implications for disease transmission or mosquito control.

Author Response

Reviewer 3 (please use the line in PDF version)

  1. Some changes have been made to the manuscript which have improved its clarity and helped the reader to understand the methods and results more easily. However, I would still like to see better framing of the study – why these were interesting questions, what the observations of histopathology and morphology mean, what the implications of the data are, and how the results might be used to better understand disease transmission or improve mosquito control. However, I would still like to see better framing of the study
    • Why these were interesting questions?

Response: There is a lack of knowledge about fine particulate matter (PM2.5) effects on Arbovirus vectors. (Line 23; highlight in blue color on Simple Summary part)

  • What the observations of histopathology and morphology mean?

Response: We are looking all tissues and morphology of the whole body of Ae. aegypti mosquitoes however we found only changing at the ovary tissues and some part of the body. We have discussed the result of tissue and morphological study in lines: 135-141, 409-411 and 393

  • What the implications of the data are, and how the results might be used to better understand disease transmission or improve mosquito control.

Response: The present study explored the effects of PM2.5 exposure on the blood feeding rate and organ integrity in the major arboviral vector Ae. Aegypti, providing information on the potential, but indirect operational impact of PM2.5 exposure on the survival and transmission capabilities of this major vector. Our findings may contribute towards the conceptualization and implementation of mosquito control measures with due consideration of the effects of ambient PM2.5 on their populations. (Lines 31-36)

The following changes have not been made in the text, even though in some cases the explanations are given in the authors’ response:

  1. Reviewer:I think that the author has said that this was a qualitative assessment of the presence of vacuolated ovaries, and so it is not appropriate to quantify the number in the way I suggested.

Response: Thank you for your kind suggestion and sorry for our mistaken. Figure 6 was rearranged following your comments. Only an example of histopathological images was showed with indicating clearer arrow. Ovarian degeneration score was added with the correlation between its severity and PM2.5 concentration. (lines 219-224 and 330-335) 

  1. Reviewer:No text has been added to the manuscript to explain why the ovaries were examined, or why the presence of particles on the mosquitoes’ surface was observed. This explanation given here would be useful in the text, along with a mention that the number of particles and vacuolated ovaries increased with PM5 concentration, and that the mosquitoes were likely under more stress as a result.

Response: Please kindly find these changes in the Materials and Methods; Histopathological study on page 4-5, and in the Results on page 10. We have state in the morphological study that we examine the whole body of Ae. aegypti mosquitoes. (Highlight in blue color on Materials and Methods; Morphological Study part) but found only abnormal part in the antenna and the ovaries. We also discuss our postulate results of morphological study in lines: 135-141, 409-411 and 393

  1. Reviewer:Some text has been added to the Introduction, but it still does not explain why these parameters were looked at, and what the implication might be.

Response: We have added ‘Thus, data on the specific association between PM2.5 pollution and the Arbovirus vector are sparse, especially data on the specific relationship between PM2.5 and the biology of a dengue vector. Thus, we aimed to explore the link between PM2.5 exposure and entomological indices of Ae. aegypti. A PM2.5 generating chamber was set up, and Ae. aegypti activity was observed. Histopathological and electron microscopy studies were performed to identify any histopathological or ultrastructural changes on the mosquitoes’ bodies.’ Into the introduction part. (lines: 135-141)

  1. Reviewer:This explanation should be added to the manuscript – it is still not clear as written that a range of concentrations were used or how this was achieved.

Response: We have added the sentence ‘The concentration of PM2.5 in an experimental chamber was observed and maintained within the set-up test level by observers who nebulized PM2.5 into an experimental chamber according to real-time PM2.5 monitoring by DustBoy machine.’ In Materials and Methods. (Line 184; after ref. [29] highlight in blue color on Materials and Methods, Environmental chambers and PM2.5 generator part)

  1. Reviewer:This should be stated in the manuscript.

Response: We have state in the morphological study that we examine the whole body of Ae. aegyptimosquitoes. (Highlight in blue color on Materials and Methods; Morphological Study part)

7.Reviewer: One sentence was added to restate what was done in the study, and I would still like to see more discussion of the possible usefulness of the data collection and implications for disease transmission or mosquito control.

Response: We have added the sentence ‘The results are important to understand blood feeding or feeding pattern of mosquito during PM2.5 pollution. This information has the potential to inform the accurate con-ceptualization and implementation of successful mosquito control programs during high level of PM2.5. Further longitudinal studies are needed to see the effect of PM2.5 on the life cycle of Ae. Aegypti and to confirmed the association between air pollution and vector-borne diseases.’ (Highlight in blue color on last part of Discussion)
